# Optimization of *Aspergillus versicolor* Culture and Aerosolization in a Murine Model of Inhalational Fungal Exposure

**DOI:** 10.3390/jof9111090

**Published:** 2023-11-08

**Authors:** Catherine B. Blackwood, Tara L. Croston, Mark A. Barnes, Angela R. Lemons, Rachael E. Rush, Travis Goldsmith, Walter G. McKinney, Stacey Anderson, Kelly L. Weaver, Michael Sulyok, Ju-Hyeong Park, Dori Germolec, Donald H. Beezhold, Brett Green

**Affiliations:** 1Allergy and Clinical Immunology Branch, Health Effects Laboratory Division, National Institute for Occupational Safety and Health, Centers for Disease Control and Prevention, Morgantown, WV 26505, USA; njz7@cdc.gov (C.B.B.);; 2Department of Pharmacology & Toxicology, Indiana University School of Medicine, Indianapolis, IN 46202, USA; 3Department of Agrobiotechnology (IFA-Tulln), Institute of Bioanalytics and Agro-Metabolomics, University of Natural Resources and Life Sciences, 1180 Vienna, Austria; 4Respiratory Health Division, National Institute for Occupational Safety and Health, Centers for Disease Control and Prevention, Morgantown, WV 26505, USA; 5Division of Translational Toxicology, National Institute of Environmental Health Sciences, Research Triangle Park, NC 27709, USA; 6Office of the Director, Health Effects Laboratory Division, National Institute for Occupational Safety and Health, Centers for Disease Control and Prevention, Morgantown, WV 26505, USA

**Keywords:** *Aspergillus*, *Aspergillus versicolor*, repeated indoor fungal exposure, allergen, aerosolization, murine nose-only exposure

## Abstract

*Aspergillus versicolor* is ubiquitous in the environment and is particularly abundant in damp indoor spaces. Exposure to *Aspergillus* species, as well as other environmental fungi, has been linked to respiratory health outcomes, including asthma, allergy, and even local or disseminated infection. However, the pulmonary immunological mechanisms associated with repeated exposure to *A. versicolor* have remained relatively uncharacterized. Here, *A. versicolor* was cultured and desiccated on rice then placed in an acoustical generator system to achieve aerosolization. Mice were challenged with titrated doses of aerosolized conidia to examine deposition, lymphoproliferative properties, and immunotoxicological response to repeated inhalation exposures. The necessary dose to induce lymphoproliferation was identified, but not infection-like pathology. Further, it was determined that the dose was able to initiate localized immune responses. The data presented in this study demonstrate an optimized and reproducible method for delivering *A. versicolor* conidia to rodents via nose-only inhalation. Additionally, the feasibility of a long-term repeated exposure study was established. This experimental protocol can be used in future studies to investigate the physiological effects of repeated pulmonary exposure to fungal conidia utilizing a practical and relevant mode of delivery. In total, these data constitute an important foundation for subsequent research in the field.

## 1. Introduction

Fungi are ubiquitous in natural and built environments and serve an important biological role in decomposition, nitrogen fixation, and cycling of other nutrients. Fungal contamination of buildings has historically been associated with natural disasters, for instance, floods and hurricanes, which are occurring at increasing rates and severity due to global climate change [1,2,3]. Despite not being considered pathogenic, chronic and repeated exposures to fungal spores and conidia have been correlated to negative health impacts. Contaminated environments such as water-damaged occupational buildings and residential homes serve as incubators that grow high quantities of fungal spores and are locations in which individuals spend large amounts of time, therefore being exposed to high quantities of microbes for extended periods of time [4]. After contamination, a building does not return to its pre-mold-infested state without significant, costly remediation [5]. As a result, individuals are chronically exposed to high concentrations of fungi, and there is an additional burden on lower-income individuals and families who cannot afford remediation. These observations have resulted in identification of “sick building syndrome”. According to a systematic review, several factors contribute to the contamination of buildings, including dampness, building components, and occupant behavior [6]. These factors, in combination with genetic predisposition of the occupants, contribute to development of respiratory disease and asthma morbidity and mortality [7,8,9]. In combination with the increased incidence of mucormycosis in some countries during the COVID-19 global pandemic, there is a great effort to understand the impact of chronic, repeated fungal exposure [10].

Built environments are typically contaminated with a combination of multiple fungal species. Several genera of fungi, including *Aspergillus*, *Penicillium*, *Cladosporium*, *Ulocladium*, *Alternaria*, and *Wallemia*, have been repeatedly correlated with health risks to asthmatic patients [6,11]. Airway exposure to these fungi has been associated with rhinitis, asthma, atopic dermatitis, allergic fungal sinusitis, and other disease manifestations through mechanisms unique to each fungal species [11,12,13]. While prevention would be the gold standard for thwarting fungi-induced health effects, and effective mitigation of fungal contamination is still a priority, understanding the mechanisms of disease development can help to identify safe levels of exposure, biomarkers of exposure and effect, and therapeutic targets for disease. The mechanistic role of each species in the induction of respiratory sensitization is unknown; thus, each species needs to be studied individually before an investigation of the milieu can be undertaken.

Fungal species within the *Aspergillus* genus are frequently identified and isolated from fungal-contaminated buildings [3,6,12,14,15,16,17,18]. *Aspergillus versicolor* is one of three commonly identified *Aspergillus* species, the other two being *Aspergillus fumigatus* and *Aspergillus niger* [19]. In this study, we focus on *A. versicolor*, which has been isolated from a variety of locations around the world, including from extreme conditions such as desert and hypersaline aquatic environments [14], damp building materials, agricultural straw [20], and soil [14]. *A. versicolor* is a species recognized as an indicator organism for building moisture problems [21]. It has become increasingly appreciated that there is a relationship between fungus and building-associated allergic respiratory diseases, though a causal or mechanistic relationship between the two has not been fully elucidated [3,15,18,22]. For these reasons, it is vital that a model for laboratory animal exposure to *A. versicolor* spores is developed, optimized, and evaluated for future mechanistic immunological studies.

Several previous studies have examined the physiological impact of exposure to *A. versicolor*, its toxins, and other metabolites [3,15,18,21,23]. A study using intratracheal instillation of *A. versicolor* demonstrated that a single dose exposure to a dose of a spore suspension was capable of inducing inflammation and toxicity in mice [15]. A subsequent investigation revealed that *A. versicolor* extracts induced a dose-responsive inflammatory response [21]. While studies that utilize liquid fungal suspensions or extracts can provide valuable insight into the impact of *A. versicolor* exposure, these methodologies do not mimic a natural exposure as does an inhalational method using dry, viable spores.

Studies examining the immunotoxicological response to *Stachybotrys chartarum* and *Aspergillus fumigatus* were previously performed by our group using a unique nose-only acoustical generator system (AGS) [24,25,26,27,28]. This system allows for exposure to dry conidia, a more accurate representation of real-world fungal exposure in comparison to intranasal or intratracheal instillation. The AGS allows for the delivery of dry conidia to live, awake animals. Previous studies performed with *A. fumigatus* were designed to identify the dose of conidia necessary to induce lymphoproliferation but not to cause infection. Repeated exposure to *A. fumigatus*, which is considered an opportunistic pathogen, led to a skewing of the immune response towards Th2 type response and increased pro-allergy IL-13, IL-4, and IgE ([27,29]). Similar studies have focused on *Stachybotrys chartarum*, a species that is not considered pathogenic and is commonly known as black mold. *S. chartarum* has also been linked to damp-building-related illnesses, including respiratory disease, allergic dysregulation, and potentially neurological symptoms [30].

In this study, the culture methods and aerosolization of viable and heat-inactivated *A. versicolor* are demonstrated for feasibility and reproducibility, and then the deposition and lymphoproliferative qualities of the spores were surveyed. Next, mice were exposed to a range of *A. versicolor* doses to determine deposition and a lymphoproliferative dose to use for subsequent exposure studies. Finally, an exposure study was conducted to examine the pulmonary responses to repeatedly inhaling *A. versicolor*.

## 2. Materials and Methods

### 2.1. Culture of Aspergillus versicolor

*Aspergillus versicolor (Vuillemin) Tiraboschi*, ATCC 9577 (American Type Culture Collection, Manassas, VA, USA) was inoculated onto Malt Extract Agar (MEA) using a sterile inoculating loop and incubated at 23 °C with 98% humidity for 7 days. Conidia were recovered from the MEA with a sterile inoculating loop in sterile distilled water and diluted to a concentration of 2.5 × 10^6^ conidia/mL in 5 mL. The suspension of conidia was then inoculated onto 10 g of autoclaved (132 °C for 30 min) Mahatma white rice (Riviana Foods Inc., Houston, TX, USA) in 100 mm Petri dishes and grown at 23 °C, 98% humidity for 7–10 days. Heat-inactivated *A. versicolor* conidia were used throughout these studies as a biological particle control. Rice cultures were baked in a laboratory oven (Thermo Fisher, Waltham, MA, USA) at 80 °C for 4 h to produce heat-inactivated *A. versicolor* conidia. Viability was tested by plating 100 viable or heat-inactivated *A. versicolor* spores on MEA plates and counting colony growth after 72 h. Viable and heat-inactivated *A. versicolor* rice cultures were desiccated in vacuum-sealed chambers for 3–5 days or 7–10 days where noted.

### 2.2. A. versicolor Protein Extract & SDS PAGE Analysis

*A. versicolor* was inoculated onto MEA as described above, and conidia were recovered from the surface of MEA plates into sterile distilled water using an inoculating loop. Viable or heat-inactivated *A. versicolor* conidia were recovered from rice grains by washing and agitation in sterile distilled water. *A. versicolor* conidia were frozen at −80 °C and then lyophilized. Lyophilized conidia were disrupted with 0.5 mm glass beads using a Bead Mill 24 Homogenizer (Thermo Fisher, Waltham, MA, USA). Conidia and beads were washed in PBS, centrifuged at 400× *g*, and protein supernatant was collected. Protein concentration was determined via Pierce BCA Assay (Thermo Fisher, Waltham, MA, USA), and 30 µg was resuspended in Laemmli buffer with β-mercaptoethanol. Suspensions were heated at 95 °C for 5 min. Samples were resolved on 4–20% polyacrylamide gels (Bio-Rad, Hercules, CA, USA) and stained with Imperial Protein Stain (Thermo Fisher, Waltham, MA, USA) according to the manufacturer’s recommendation. Protein distribution was compared.

### 2.3. Microbial Metabolite Quantification

*A. versicolor* was cultured on rice, as described above. Following desiccation, spore-laden rice was mixed with PBS and viable or heat-inactivated conidia were liberated and collected, then quantified using a hemocytometer. Aliquots of 5 × 10^5^ cells were lyophilized. Samples were shipped on dry ice and stored frozen until analysis by mass spectroscopy. The lyophilized spores were extracted for 90 min with 200 µL of acetonitrile/water/acetic acid in a 79/20/1 ratio.

Detection and quantification were performed with a QTrap 5500 MS/MS system (Applied Biosystems, Foster City, CA, USA) equipped with a TurboIonSpray electrospray ionization (ESI) source and a 1290 series UHPLC system (Agilent Technologies, Waldbronn, Germany). Chromatographic separation was performed at 25 °C on a Gemini^®^ C18-column, 150 × 4.6 mm i.d., 5 μm particle size, equipped with a C18 security guard cartridge, 4 × 3 mm i.d. (all from Phenomenex, Torrance, CA, USA. ESI-MS/MS was performed in the scheduled multiple reaction monitoring (sMRM) mode, both in positive and negative polarities, in two separate chromatographic runs. The sMRM detection window of each analyte was set to the respective retention time ±27 s and ±42 s in positive and in negative mode, respectively. The target scan time was set to 1 s. The settings of the ESI-source were as follows: source temperature 550 °C, curtain gas 30 psi (206.8 kPa of max. 99.5% nitrogen), ion source gas 1 (sheath gas) 80 psi (551.6 kPa of nitrogen), ion source gas 2 (drying gas) 80 psi (551.6 kPa of nitrogen), ion-spray voltage −4500 V and +5500 V, respectively, collision gas (nitrogen) medium. The preparations were screened for a total of 662 analytes. One preparation contained traces of *Penicillium* metabolites and was therefore excluded from analysis.

### 2.4. Sterigmatocystin Quantification

Single grains of conidia-laden rice were weighed to acquire the mass. Forty nanograms of isotopically labeled ^13^C sterigmatocystin (as internal standard) in acetonitrile was spiked onto non-inoculated grains, and acetonitrile was allowed to evaporate. One milliliter of extraction solution (a combination of 99% acetonitrile and 1% acetic acid by volume) was then added into a glass tube with the grain. Sterigmatocystin was extracted from fungal rice cultures by shaking for 90 min in an extraction solution. After shaking, the tube was centrifuged for 3 min at 1962× *g* 700 µL of the supernatant and was transferred to an injection vial that was then dried under a gentle nitrogen stream. Once dried, 800 µL of injection solution (a combination of 69% water, 30% methanol, and 1% acetic acid by volume) was added to the injection vial and vortexed for 30 s. Finally, 10 µL of the sample extract was injected into an Ultra-Performance Liquid Chromatograph coupled with a Tandem Mass Spectrometer (UPLC/MSMS, Acquity H Class UPLC and Acquity Xevo TqD Quadrupole Tandem Mass-Spectrometer, Waters, MA, USA). The gradient mobile phase, starting with 90% aqueous and 10% organic, was pumped at a flow rate of 0.2 mL/min when sample extract was injected. The aqueous solvent with 10 mM of ammonium acetate was composed of 99% super pure water and 1% acetic acid (by volume), and the organic solvent with 10 mM ammonium acetate was composed of 99% methanol and 1% acetic acid (by volume). The percentages of the mobile phase were changed over time to allow the organic solvent to be dominant (97%) at 5.7 min so that the sterigmatocystin was successfully eluted. Extraction of sterigmatocystin from air filters used the same procedure with the exception that 200 µL of injection solution was used to reconstitute dried extract prior to identification using UPLC/MSMS. Sterigmatocystin (molecular weight 324.1 g/mol) was identified and quantified with a precursor ion (325.159 Dalton (Da), [M+H]^+^) and two product ions with molecular weights of 310.086 and 280.964 Da at collision energy of 30 and 26 kV, respectively. The cone voltage suitable for this ionization was 50 kV.

### 2.5. Endotoxin Quantification

The endotoxin potency (EU—endotoxin unit) of each sample was determined using the kinetic chromogenic *Limulus* amoebocyte lysate (LAL; Associates of Cape Cod, Inc., Falmouth, MA, USA) assay method. Reference endotoxin (lot # HOK354, *E. coli* O113:H10 strain) was purchased from the US Pharmacopeia (Rockville, MD, USA). All glassware used for the assay was baked at 260 °C for 2–2.5 h to remove endotoxins, and a sterile technique was applied to prevent cross-contamination during the assay. Prior to the assay, 46.7 to 58.0 mg of conidia-laden rice grains or filters were extracted in 2.5 mL or 3.0 mL of triethylamine phosphate buffer, respectively, by sonicating in an ultrasonic water bath sonicator (Bransonic^®^ Model 5510, Branson Ultrasonics Corp., Danbury, CT, USA) for 1 h. Serial dilutions were prepared for all sample extracts and standards in LAL reagent water (LRW). A 50 µL aliquot of each of the serial dilutions and negative controls was loaded onto a 96-well microplate in duplicate, and 50 µL of LAL reconstituted with Glucashield^®^ buffer (Associates of Cape Cod, Inc., Falmouth, MA, USA) was added to all standards, samples, and negative controls. Immediately after loading, the microplate was placed in a pre-incubated plate reader (ELx808IU™, Biotek Instruments, Inc., Winooski, VT, USA) at 37 °C and shaken for 30 s. The optical density (OD) was recorded at 405 nm every 30 s for 120 min. The onset time was defined for each well as the time when the OD reading for the well exceeded and remained higher than the threshold (OD = 0.02). A parallel-line estimation method [31] was used to estimate endotoxin potency relative to reference standard endotoxin for each sample.

### 2.6. Conidia Aerosolization

The inhalation exposure system used in these studies was previously adapted from the Pitt 3 generator and utilized by NIOSH to allow nose-only exposure of mice to fungal conidia [24,32,33]. Clumps of *A. versicolor*-covered rice grains were broken apart by hand in an antistatic bag (LJY, Brighton, CO, USA) and placed in an acoustical particle generator. The acoustical generator system (AGS) produced sound energy to vibrate the conidia-laden-rice inside a polycarbonate tube (12-inch diameter × 18-inch height) that had both ends capped with thin latex rubber. Vibrations were controlled by custom software that delivered a 15 Hz sine wave to an amplifier. This resulted in the aerosolization of the conidia of the rice, which was then mixed with air (7 L/min) and passed into the nose-only exposure chamber and pods. An empty nose-only exposure pod was used to collect samples onto 25 mm 0.2 µm pore size Whatman polycarbonate filters (Whatman/GE Healthcare, Chicago, IL, USA) for field emission electron microscopy, mycotoxin quantification, as well as to determine the aerodynamic size of the particles (APS, TSI, Inc., Shoreview, MN, USA). Additionally, a light scattering device, DataRAM4 (ThermoElectron Co., Franklin, MA, USA), was used to provide a real-time estimate of the mass concentration of *A. versicolor* conidia within the chamber. Gravimetric determinations (37 mm cassettes with 0.45 µm pore-size Teflon filters, 1.5 L/min. sample flow) were used to calibrate and verify the Data RAM4 readings during each exposure run. The temperature and humidity were monitored and recorded in real time during each exposure using a probe (HMP60, Vaisala Inc., Louisville, CA, USA) that was adapted to an empty mouse pod. Images of conidia recovered from filters were acquired using a Hitachi S-4800 Field Emission Scanning Electron Microscope (Hitachi, Tokyo, Japan).

### 2.7. Murine Nose-Only Inhalation Exposures

Five- to six-week-old female C57BL/6 mice (Jackson Laboratory, Bar Harbor, ME, USA) were housed in HEPA-filtered plastic cages. Mice had *ad libitum* access to Rodent NTP-2000 diet (Zeigler Bros., Inc., Gardners, PA, USA) and water. All procedures performed on animals were approved by the CDC-Morgantown Animal Care and Use Committee. Several experimental designs were utilized in this manuscript, as described below and indicated in the corresponding result text. Quantifications of inhaled spores represent the administered dose, not exposure levels unless otherwise noted.

#### 2.7.1. Conidia Deposition Studies

C57BL/6 mice were exposed to either HEPA-filtered air, viable *A. versicolor* conidia resulting in an estimated (1 × 10^5^), or heat-inactivated *A. versicolor* conidia (3.4 × 10^4^) spores deposited in mouse lungs over a single 60 min duration exposure. The animals were euthanized immediately following the completion of the exposure via the administration of a final dose of 100–300 mg/kg pentobarbital via intraperitoneal injection. As a secondary method of euthanasia, exsanguination was performed, and blood was collected via a cardiac puncture of the right ventricle and placed in serum collection tubes. The lungs were inflated with neutral buffered formalin using a catheter inserted into the trachea, tied off, and the lung was removed with the trachea intact. The tissue was then placed in a 15 mL tube containing 10 mL of 10% formalin for 24–48 h. Whole heads were also fixed in formalin for examination of the nasal passage.

#### 2.7.2. Modified Murine Local Lymph Node Assay (LLNA)

For the modified local lymph node assay, five groups of C57BL/6 mice were exposed to either HEPA-filtered air, viable *A. fumigatus* conidia (1 × 10^5^), or viable *A. versicolor* conidia (1 × 10^4^, 1 × 10^5^, 5 × 10^5^) for 60 min for three consecutive days, followed by two days with no exposure. Mice were then injected intravenously via the lateral tail vein with 20 µCi 3H-thymidine (Dupont NEN, Waltham, MA, USA; specific activity 2 Ci/mmol) 48 h following the final exposure, then euthanized via CO_2_ inhalation after five hours. The mediastinal lymph nodes were removed homogenized between frosted microscope slides, and cell suspensions were prepared. Samples were incubated with 5% trichloroacetic acid overnight at 4 °C and then mixed with scintillation fluid and counted using a Tri-Carb 2500TR liquid scintillation counter (PerkinElmer, Waltham, MA, USA). Stimulation indices (SIs) were calculated by dividing the mean disintegrations per minute (DPM) of the conidia-exposed mice by the mean DPM obtained from the air-only control mice. Concentrations of *A. versicolor* that elicited a stimulation index (SI) greater than 3 were selected for sub-chronic exposures.

#### 2.7.3. Dosimetry Studies

For the 4-week dosimetry studies, groups of C57BL/6 mice were exposed twice weekly for a total of eight exposures, to doses per exposure of viable *A. versicolor* or HEPA-filtered air. The per-exposure doses included were 1 × 10^4^, 3 × 10^4^, 1 × 10^5^, or 5 × 10^5^ number of conidia deposited in the lung. Twenty-four hours following the final exposure, animals were humanely euthanized and dissected, as described below. Mice were weighed and euthanized using an administered dose of 100–300 mg/kg pentobarbital euthanasia solution via intraperitoneal injection. As a secondary method of euthanasia, exsanguination was performed, and blood was collected via cardiac puncture of the right ventricle and placed in serum collection tubes. Following blood collection, tissues were collected for either histology or flow cytometry. In the mice used for histology, the right lung was inflated with 1 mL neutral buffered formalin using a catheter inserted into the trachea, tied off, and the lung was removed with the trachea intact. The tissue was then placed in a 15 mL tube containing 10 mL of 10% formalin for 24–48 h. For flow cytometric analysis, bronchioalveolar lavage and the lungs were collected. The bronchioalveolar lavage fluid (BAL) was collected by making a small nick in the trachea and using a catheter to wash the 2 mL total sterile PBS. The lung was then removed for flow cytometry processing, as described below.

### 2.8. Histopathology

Right lungs from control and *A. versicolor* exposed mice were formalin-fixed and submitted to HistoTox Labs (Boulder, CO, USA) for routine processing. Each lung was paraffin-embedded in a single block and one slide per block was sectioned and stained with hematoxylin and eosin (H&E). Glass slides were evaluated by a board-certified veterinary pathologist using light microscopy. Images were captured using an Olympus BX41 microscope with an Olympus SC180 camera. Lesions were given a severity score of 0–5 (0 = not present/normal, 1 = minimal, 2 = mild, 3 = moderate, 4 = marked, 5 = severe) [34].

### 2.9. Quantifcation of Conidia in Airway

At necropsy, lungs were fixed by instillation with 10% neutral-buffered formalin, processed into paraffin blocks and sectioned at 5 microns. Lung sections and nasal passages were stained with Grocott’s methenamine silver stain (GMS) for evaluating fungal deposition. GMS images were captured using an Olympus BX53 microscope in conjunction with an Olympus DP73 camera.

### 2.10. Serum Immunoglobulin Quantification

Serum was collected from whole blood following cardiac puncture. Serum concentrations of total IgG, IgA, IgM, and IgE were quantified using Invitrogen ELISA kits, per the manufacturer’s instructions (Thermo Fisher, Waltham, MA, USA).

### 2.11. Flow Cytometry

Whole lungs from control- and *A. versicolor*-exposed mice were minced and digested in DMEM (Thermo Fisher, Waltham, MA, USA) with 5% Fetal bovine serum (Thermo Fisher, Waltham, MA, USA), 1 mg/mL Collagenase/Dispase (Roche Diagnostics, Mannheim, Germany), and 30 µg/mL Deoxyribonuclease I (Thermo Fisher, Waltham, MA, USA) for 25 min at 37 °C. Following digestion, a single-cell suspension was obtained, and cells were incubated with normal rat sera (MilliporeSigma, Darmstadt, Germany) and stained using rat anti-mouse CD16/32 (clone 2.4G2, BD Biosciences, San Jose, CA, USA) in PBS for 5 min at room temperature. Cells were then stained for 25 min at 4 °C with an antibody cocktail containing PBS, 5% BSA, 2 mM EDTA and the following fluorochrome-conjugated antibodies: CD11b (clone M1/70), CD11c (clone N418), Ly6C (HK1.4), Ly6G (1A8), CD49b (clone DX5), CD3 (clone 17A2) (Biolegend, San Diego, CA, USA), Siglec F (clone E50-2440), CD103 (clone M290), CD19 (clone 1D3) (BD Biosciences, San Jose, CA, USA). The antibody staining cocktail was washed away with FACS buffer. Cells were centrifuged (400× *g*, 5 min at 4 °C), then fixed in the BD Cytofix Fixation Buffer (BD Biosciences, San Jose, CA, USA) for 10 min at room temperature and washed with FACS buffer and centrifuged. Cells were resuspended in FACS buffer, and data were acquired using an LSR II flow cytometer (BD Biosciences, San Jose, CA, USA). Data were analyzed using FlowJo software (version 10.6.2; FlowJo LLC, Ashland, OR, USA).

### 2.12. Statistics and Data Representation

All statistics, unless otherwise noted, were performed in GraphPad Prism for Windows (version 9.3.0, GraphPad Software, San Diego, CA, USA). Analyses are described in the text as they correspond to the data represented.

## 3. Results

### 3.1. A. versicolor Culture and Heat-Inactivation

Prior to their use in murine-exposure studies, the growth and inactivation of *A. versicolor* conidia were optimized. To test the viability of spores grown on rice following desiccation, 100 conidia were counted and plated on MEA. After 72 h, viable colonies were counted, and the percent viable was calculated (Figure 1A). Following the optimized growing condition, an average of 78% of conidia were viable and produced colonies. Heat treatment resulted in a significant decrease in viability, and no heat-inactivated conidia produced viable colonies (*t*-test, *p* < 0.0001; Figure 1A). Additionally, field emission scanning electron micrographs showed that the live and heat-inactivated conidia exhibited similar morphological properties (Figure 1B,C) and were aerosolized as single conidia, as well as in conidia aggregates with little to no conidial or hyphal fragmentation. These data illustrate that the *A. versicolor* is culturable under these conditions and that the heat treatment used is sufficient to generate a heat-inactivated particle control for exposure studies.

### 3.2. Aerosolization of Viable and Heat-Inactivated A. versicolor Conidia Using AGS

In this study, mice were exposed in nose-only chambers to *A. versicolor* conidia. To aerosolize the conidia and free them from the rice on which they were grown, the conidia must be desiccated. Here, a desiccation interval of five days was compared to a desiccation interval of seven days. Regardless of desiccation or heat inactivation, the *A. versicolor* conidia released from rice during aerosolization were, on average, 2.5 µm in diameter (Figure 1D–F). Longer desiccation did not impact the size distribution of the conidia (Figure 1E). At similar levels of input power to the AGS, more conidia (4–5×) were released from the rice with five days of desiccation as opposed to seven days. Given these data, a desiccation period of five days was selected for subsequent studies.

### 3.3. Viable and Heat-Inactivated Conidia Characterization

Given that viable and heat-inactivated conidia were morphologically similar and able to be aerosolized in similar conditions, the comparability on a molecular level was also of interest. To assess this, preparations of viable and heat-inactivated *A. versicolor* were lyophilized, and then protein was extracted, or secondary metabolites were assessed via mass spectrometry. To examine the protein profile, 30 µg of each preparation was run on a 4–20% SDS-PAGE, and the band profiles were compared (Figure 2A). Banding patterns in each preparation were similar, suggesting that heat inactivation did not cause significant disruption of the cellular architecture. Next, secondary metabolites of the fungi were quantified, and the amounts found in preparations of viable and heat-inactivated conidia were compared. There was not a significant change in the amount of sterigmatocystin, aspercolorin, averantin, averufin, nidurufin, norsolorinic acid, seco-sterigmatocystin, versicolorin A, versicolorin C, or versiconal acetate (Figure 2B). Of note, there was a significant decrease in the amount of versiconol in the heat-inactivated conidia compared to viable (Figure 2B). Next, we examined the quantities of sterigmatocystin delivered to the mice during exposure (Figure 2C). To accomplish this, air filters were placed in exposure pods, collected, and later examined. The air-only control (AOC) filter had no detectable levels of sterigmatocystin, but the viable and HIC *A. versicolor* filters did. There was a significant decline in the sterigmatocystin in the HIC *A. versicolor* group compared to the viable *A. versicolor* (Figure 2C). Finally, the amounts of endotoxin present in the rice and the filter were quantified using LAL (Figure 2D). While there were low levels present in the rice, the levels were reduced on the viable *A. versicolor* cultured rice and further decreased on the HIC *A. versicolor* rice grains (Figure 2D). Additionally, no endotoxin was detected on the filters that collected filtrate from the AGS (Figure 2D).

### 3.4. A. versicolor Conidia Are Deposited into Lung following Nose-Only Exposure

Following optimization of growth and aerosolization of *A. versicolor* conidia, deposition studies were performed to quantify the deposition of the viable and heat-inactivated conidia into the airways of mice via nose-only exposure. Groups of C57BL/6 mice were exposed to HEPA-filtered air control, viable *A. versicolor* conidia (1 × 10^5^), or heat-inactivated *A. versicolor* conidia (3.4 × 10^4^) during a single 60 min exposure. The nasal passages and lungs of mice were examined by histopathology for deposition of conidia, using GMS. Both the viable and heat-inactivated conidia were identified in the nares and were observed aggregating at comparable levels on the epithelium of the nose (Figure 3A). Viable *A. versicolor* conidia were observed at similar levels in the lung, but significantly fewer heat-inactivated *A. versicolor* conidia were deposited in the lung (Figure 3A). Of note, the non-viable conidia found in the nasal passage also appeared clustered, whereas the viable conidia in the nasal passage were more separated (Figure 3B).

Taken together, THE data demonstrated that the *A. versicolor* culture method used in these studies resulted in aerosolized viable *A. versicolor* conidia, which produced the mycotoxin sterigmatocystin and deposited into the lungs of mice using the AGS. We next sought to optimize the murine exposure paradigm, including THE identification of a viable spore dose that induced lymphoproliferation and immunoreactivity.

### 3.5. Identifying the Lymphoproliferative Dose of A. versicolor

To determine the dosage necessary to induce an immune response following respiratory exposure to live *A. versicolor* conidia, two experimental approaches were utilized. First, a modified LLNA was performed to identify the optimal dose of *A. versicolor* conidia at which lymphoproliferation was observed. Groups of C57BL/6 mice were exposed to either HEPA-filtered air, viable *A. versicolor* conidia (1 × 10^4^, 1 × 10^5^, 5 × 10^5^), or viable *A. fumigatus* conidia (1 × 10^5^) for 60 min for three consecutive days, followed by two days of recovery. *A. fumigatus* was selected as a positive control for this study because previous data demonstrated strong immunogenicity of *A. fumigatus* conidia in this model of exposure [24,25]. Mice then received an IV tail injection of 20 µCi 3H-thymidine and were euthanized five hours later. Mediastinal lymph nodes were assessed using the modified LLNA, and SIs were calculated. Concentrations of *A. versicolor* that elicited a SI greater than 3 were selected for sub-chronic exposures. The air-only control (AOC) group demonstrated negligible lymphoproliferation, while lymphoproliferation increased stepwise with increasing doses of *A. versicolor* and was highest in the *A. fumigatus* positive control group (Figure 4). The low dose of *A. versicolor*, 1 × 10^4^ conidia, did not result in a significant increase in DPM in comparison to the AOC, nor was the SI higher than the selected value of three (Figure 4). However, each of the higher doses of *A. versicolor*, 1 × 10^5^ and 5 × 10^5^ conidia yielded significant increases in DPM and SIs higher than three, 8.0 and 11.9, respectively (Figure 4). Because both the 1 × 10^5^ and 5 × 10^5^ conidia doses yielded lymphoproliferation in a draining lymph node, any dosage between those values was deemed appropriate for murine exposures. Based on these data, and in combination with the reproducibility of the AGS to reach the specific spore concentration within one hour , the maximum dose of 3 × 10^5^
*A. versicolor* conidia was selected for subsequent murine exposure studies.

### 3.6. Repeated Exposure to A. versicolor Conidia

After optimizing growth, desiccation, and aerosolization of *A. versicolor* conidia and identifying a lymphoproliferative dose, subsequent experiments were designed to assess the pathological and immunological implications of repeated exposure. Groups of mice were exposed to 1 × 10^4^, 3 × 10^4^, 1 × 10^5^, or 3 × 10^5^ viable *A. versicolor* conidia or to HEPA-filtered air in nose-only exposure chambers. Mice were weighed weekly and then euthanized 24 h following the final exposure. The lungs of each mouse were examined by either histology or flow cytometry (Figure 5A). The number of deposited conidia was quantified and consistent for each target dose. For one exposure each in the 1 × 10^5^ and 3 × 10^5^ doses, the target dose was not achieved (indicated by the red inverted triangle). As a result, these exposure groups were administered an additional exposure. None of the doses of viable *A. versicolor* caused significant weight loss in the mice, indicating that the mice were not experiencing infection (Figure 5C). This is consistent with the aim of the exposure paradigm to induce a non-pathogenic but immunomodulatory exposure.

Because previous data demonstrated successful deposition of live conidia into the lung following a single exposure, lung histopathology following repeated exposure was examined. The expected histologic findings for *Aspergillus* inhalation, including fibrosis, inflammation, cellular hyperplasia/metaplasia, and vascular injury, were not observed in this study for any exposure group. Observed lesions were minimal, sporadic, and considered unlikely to be related to the experimental exposure (Appendix A).

Populations of innate immune cells, including alveolar macrophages, neutrophils, and eosinophils, were quantified in the bronchoalveolar lavage fluid and lung. Twenty-four hours after the final exposure, flow cytometry was performed on single-cell suspensions from each tissue. Following 4 weeks of exposure, we did not observe a significant increase of lymphocytes in the airways (Figure 6). Interestingly, in the mice exposed to the highest dose of *A. versicolor* conidia, there was a notable shift in the type of innate immune cells that had localized. There was a significant increase in neutrophils in the BAL and lung and a decrease in alveolar macrophages in the BAL compared to the mice that had received air-only exposure (Figure 6B,C,G). Of note, we did not observe a significant change in eosinophil populations at 4 weeks post-exposure, but there was a trend towards increased eosinophils in the BAL and lung (Figure 6D,H).

Finally, serum antibody responses following exposure to *A. versicolor* were evaluated. After four weeks, there were no significant changes in total, non-specific IgM, IgG, IgA or IgE antibody concentrations in the blood (Figure 7).

## 4. Discussion

These studies provide a foundation for future research to understand the health effects of long-term, sub-chronic fungal exposure. First, the optimal methodology for culture, heat inactivation, and aerosolization of *A. versicolor* conidia was determined. The *A. versicolor* showed high levels of viability after culture, and the methodology for inactivation was very efficient. Conidia were aerosolized and deposited into the airways of mice, induced lymphoproliferation and immune cell recruitment, but did not result in an infectious-disease response. The optimal desiccation time was determined to be 5 days. Compared to other fungi found in water-damaged environments, such as *S. chartarum*, *A. versicolor* remains more readily aerosolized [35].

It should be kept in mind that the viability of conidia grown in less-than-optimal conditions, including homes, buildings, or outdoor environments, may be lower. For this reason, it will be crucial to continue studying the impact of exposure to both live and heat-inactivated conidia. Additionally, altered growth conditions, including humidity, temperature, and time, may impact the toxins and other immunostimulatory molecules present and their quantities. One study evaluated air samples of homes with moisture damage and showed that conidia can be collected from air as single particles or in chains mixed with conidia of other mold species [36]. An in vitro study demonstrated that *A. versicolor* conidia can act in concert with *S. chartarum* conidia to elicit synergistic immune responses [37]. Inhalation of fungal particles is the natural route of fungal infection, and in the case of *A. versicolor*, inhalation of mycotoxin seems to be more toxic than other routes of exposure [38,39]. Future pulmonary mold exposure studies should utilize aerosolization methods that incorporate both single- and mixed-species cultures to evaluate health effects.

Interestingly, heat inactivation did not substantially change the amount of sterigmatocystin in this study, but in future studies, it may be of interest to investigate further if the immunological impact of heat-inactivated conidia is comparable to viable conidia. Heat-inactivated conidia exhibited similar surface morphology and would likely require exposure to a much higher temperature to change cell surface structures. Most of the conidia were aerosolized as single particles; however, some were deposited onto the collection filters as aggregates. This likely contributed to the aggregation of conidia observed in the nasal passage, as particles greater than 10 µm are generally deposited in and irritate the upper respiratory tract and nasal cavity as opposed to the lung, into which particles less than 1 µm can travel [40]. Very few hyphal and conidial fragments were detected; still, increasing the air velocity or age of the *A. versicolor* cultures may have increased the number of fragments in the bioaerosol that were delivered to the nose-only inhalation pods [41]. This is important as previous studies have reported the following: mycotoxins are associated with particles the size of conidia and hyphal fragments [35]; β-glucan is primarily derived from fungal hyphae, sterigmatocystin is detectable but not abundant in hyphae [42]; and submicron fragments are responsible for cellular toxicity [43]. These data suggest that the immunotoxicological effects observed in this study were likely driven by conidia and derived proteins and secondary metabolites such as sterigmatocystin, as opposed to other mold species, such as *A. fumigatus*, in which the physiological responses are in part driven by the presence of germinating conidia in the respiratory tract [29]. In line with these findings, the lymphoproliferative levels of even the highest dose of *A. versicolor* were not as immunomodulatory as the *A. fumigatus* group. In the future, an interesting area of study may be an investigation of the specific fungal components eliciting the observed responses.

Next, we examined the immune responses after four weeks of exposure. At this time point, it was expected that alterations to innate immune cells, including macrophages, neutrophils, and eosinophils, were occurring. Interestingly, alveolar macrophages appeared to be replaced with neutrophils and eosinophils in the BAL and lung [44,45,46]. This is consistent with the early phases of the development of an allergy-like response [47,48,49,50].

These findings present an opportunity for an interesting comparison of the interactions between fungal species and mouse strains. For example, prior studies have used the AGS to perform inhalation exposures with similar timelines, using *A. fumigatus* and *S. chartarum*, in B6C3F1/N and BALB/c mice [24,25,26]. Here, we used *A. fumigatus* as a positive control for lymphoproliferation in the modified local lymph node assay because of previous data using *A. fumigatus* in BALB/c mice [24]. Interestingly, in four-week studies of *A. fumigatus*, there were increases in macrophages, granulocytes, and lymphocytes in the BAL and lymph nodes, but that was not observed in this study [24]. *A. fumigatus* germination was also determined as an influencing factor, but in this study, there was no observed germination of the conidia in the lung. A comparison of specific virulence factors made by each species could provide insight into differences in virulence between them. Studies examining *S. chartarum* revealed that fragmentation of the spores influenced the response, but here, there was no observed fragmentation [28]. It is not fully understood how the combination of multiple species, as happens in contaminated buildings, would influence the overall response, nor is it well understood how these species would compare in side-by-side studies.

At the time point of four weeks of exposure, there were no significant changes to the non-specific serological response. In future studies, exposure paradigms longer than four weeks may allow for more investigation into the specific humoral response to fungal exposure. Not only may there be changes to the total quantity, but it may be possible to investigate antigen-specific responses, the development of IgE antibody-secreting plasma cells, and memory B cells [51,52]. Many allergic-disease responses involve the production of antigen-specific immunoglobulins [53,54]. IgE antibodies that bind allergen epitopes regulate downstream allergen-specific immune responses by enhancing mast cell activation, basophil recruitment and activation, and IgE+ B cell involvement [55]. In future studies, understanding the antigen-specific response, including binding of different antibody classes, recognition by T cells, and interaction with innate lymphoid cells, could lead to valuable insights into the mechanisms driving the response and potential routes towards mitigation [54,56,57,58].

This study highlights optimized culture and aerosolization methods that can be used to assess the pulmonary health effects of murine respiratory challenge with *A. versicolor*. Yet, there are still knowledge gaps in pulmonary cellular mechanisms, and systemic effects following exposure to *A. versicolor* and its derivatives, including mycotoxins and microbial volatile organic compounds. The need to study the acute and chronic effects of pulmonary exposure to *A. versicolor* remains vitally important [16,18]. Additionally, the ability of organisms to recover or return to a pre-exposure immunological state is unknown. Therefore, future studies should utilize a conidia aerosolization and delivery method such as the one described herein to explore the impact of fungal exposure over time across fungal species and murine strains, focusing on the mechanisms by which the responses develop.

## Figures and Tables

**Figure 1 jof-09-01090-f001:**
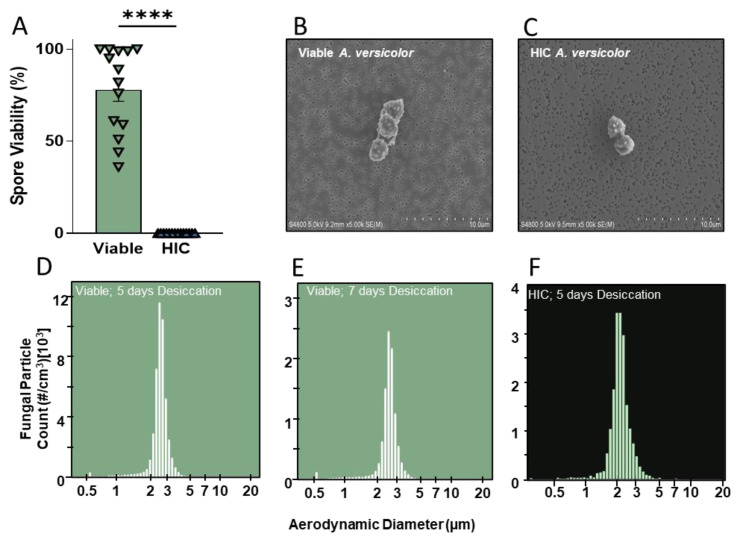
Culture viability, morphology, and aerosolization of *Aspergillus versicolor* conidia. (**A**) *A. versicolor* was grown on sterile autoclave rice. Aliquots were tested for viability before and after heat inactivation. Asterisks represent statistical significance determined by *t*-test. **** *p* < 0.0001. n = 14/treatment. (**B**,**C**) *A. versicolor* conidia were aerosolized and examined via SEM. (**D**–**F**) Mass concentration of aerosolized spores. HIC—heat inactivated conidia.

**Figure 2 jof-09-01090-f002:**
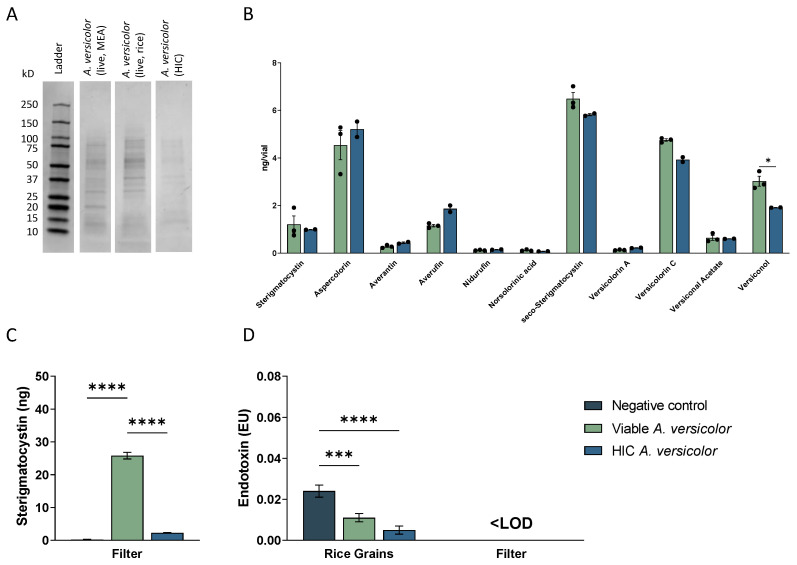
Components of viable and heat-inactivated *A. versicolor* spores. (**A**) Whole protein lysates of live *A. versicolor* collected from malt extract agar (lane 2), rice (lane 3), and heat-inactivated conidia liberated from rice (lane 4). (**B**) Absolute values of secondary metabolites associated with *A. versicolor*, quantified from live (green) and inactivated (blue) conidia. Sterigmatocystin, aspercolorin, averantin, averufin, nidurufin, norsolorinic acid, seco-sterigmatocystin, versicolorin A, versicolorin C, versiconal acetate, and versiconol were quantified from aliquots of equivalent concentrations of cells (values are represented as ng/vial of 5 × 10^5^ spores). (**C**) Sterigmatocystin quantified from filters on the receiving end of the AGS. The negative control for this figure was a filter from an AOC pod. (**D**) Endotoxin quantified on rice and from filters on the receiving end of the AGS. The negative control for the rice grain section was sterile rice, and for the filter section was a filter from an AOC pod. Asterisks represent statistical significance, as determined by two-way ANOVA, comparing cell means within rows. * *p* < 0.05, *** *p* < 0.001, **** *p* < 0.0001; n = 3 per column.

**Figure 3 jof-09-01090-f003:**
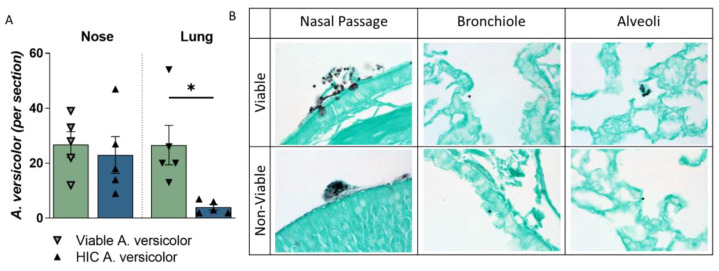
Deposition of viable and heat-inactivated spores, via AGS, into the airways of mice. A single dose of *A. versicolor* was administered to live, awake mice via an acoustical generator system. Immediately following exposure, mice were euthanized, and the lungs were assessed using histopathology. (**A**) Conidia were quantified in the nose and lung of each mouse. Asterisks represent statistical significance determined by one-way ANOVA. * *p* < 0.05 n = 5/group (**B**) Representative GMS images of each exposure group focused on the nasal passage, alveoli, and bronchiole.

**Figure 4 jof-09-01090-f004:**
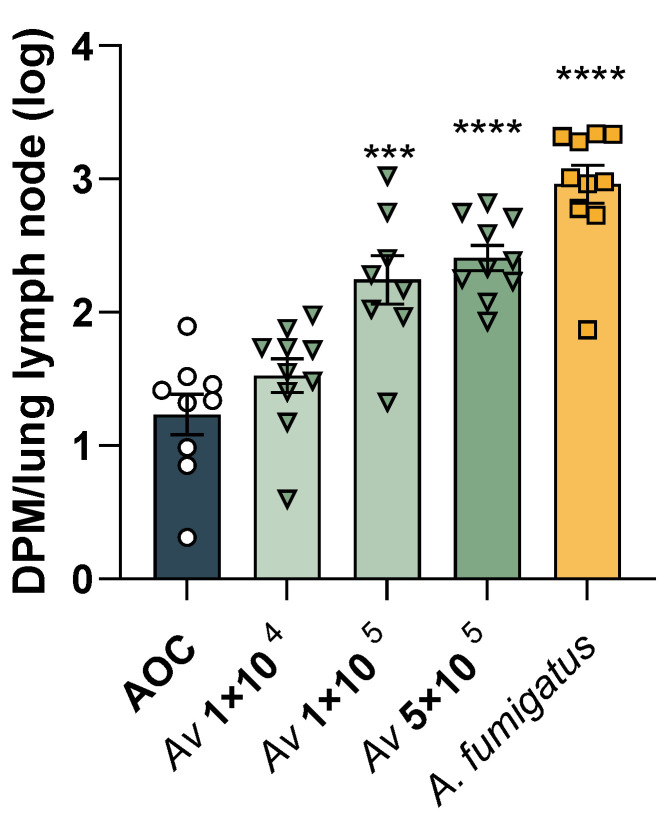
*Aspergillus versicolor* induces lymphoproliferation. A modified local lymph node assay was performed to identify lymphoproliferative doses of *A. versicolor*. A. fumigatus was used as a positive control, as it is known to be lymphoproliferative from prior studies. Values over the bars indicate the calculated SI value. Asterisks represent statistical significance, determined by one-way ANOVA. Asterisks over the bars indicate a comparison to the air-only control group. *** *p* < 0.001, **** *p* < 0.0001; n = 9–10 /group.

**Figure 5 jof-09-01090-f005:**
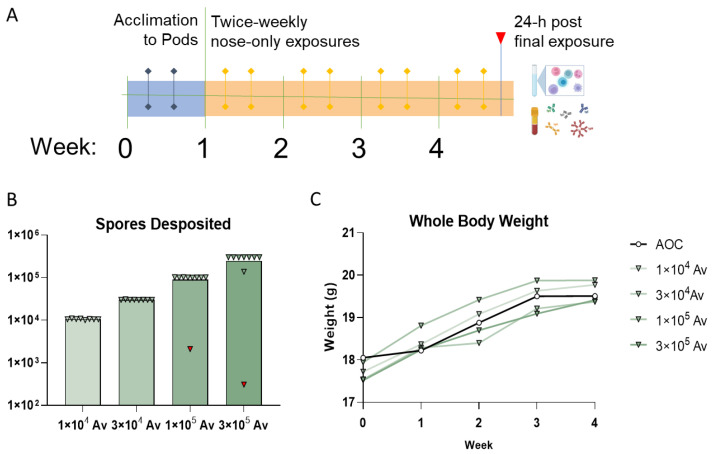
Four-week, repeated exposure murine dosimetry study. (**A**) Schematic of exposure paradigm. After a week, including two acclimations to the exposure pods, mice were exposed twice weekly for a total of eight exposures. One day after the following exposure, mice were euthanized, and samples were collected. (**B**) Conidia were quantified during each exposure and compiled for each group. Red, inverted triangles indicate aborted exposures that were later replaced. n = 8–9/group (**C**) Mouse body weight was recorded weekly to ensure animals were continuing to grow at the time of euthanasia. n = 5/group.

**Figure 6 jof-09-01090-f006:**
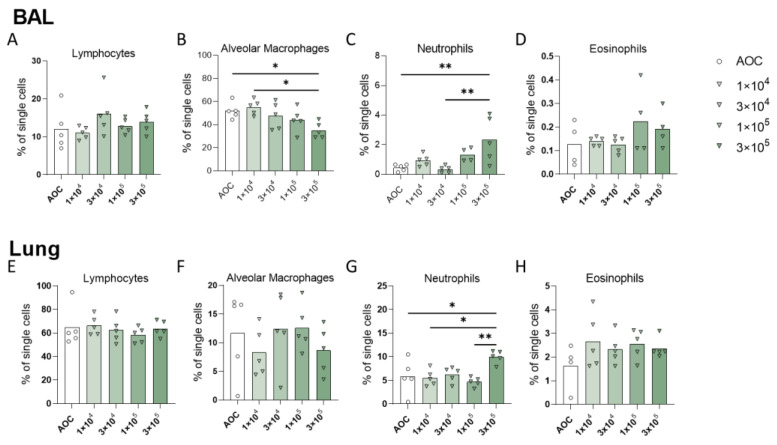
Flow cytometric analysis of cells infiltrating the lung following repeated exposure to *A. versicolor*. Flow cytometry was performed on cells from the bronchoalveolar lavage or the lung one-day post-final exposure to increasing doses of *A. versicolor*. Lymphocytes (**A**,**E**), alveolar macrophages (**B**,**F**), neutrophils (**C**,**G**) and eosinophils (**D**,**H**) were quantified as percent of single cells. Asterisks represent statistical significance as determined by one-way ANOVA with multiple comparisons. * *p* < 0.05, ** *p* < 0.01; n = 5/group.

**Figure 7 jof-09-01090-f007:**
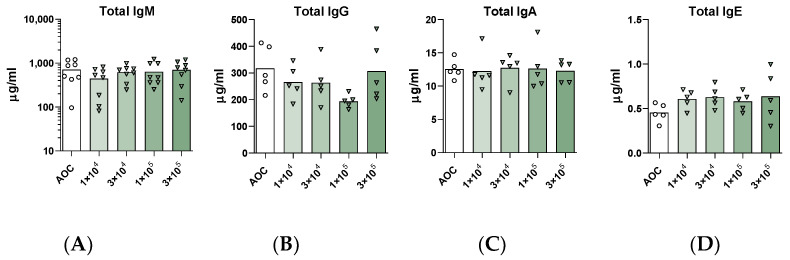
Serum immunoglobulin quantification following a 4-week dosimetry study. Total concentrations of IgM ((**A**), n = 8/group), IgG ((**B**), n = 5/group), IgA ((**C**), n = 5/group), and IgE ((**D**), n = 5/group) were quantified in serum. Statistical significance was determined by one-way ANOVA with multiple comparisons, but no significant differences were observed.

## Data Availability

All study data will be made available on the NIOSH Data and Statistics Gateway.

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
