# Peer review of "Optimization of Aspergillus versicolor Culture and Aerosolization in a Murine Model of Inhalational Fungal Exposure"

_jof, 2023, doi:10.3390/jof9111090_

Round 1

Reviewer 1 Report

Comments and Suggestions for Authors

The manuscript “Optimization of Aspergillus versicolor culture and aerosolization in a murine model of inhalational fungal exposure” is a well-planned, structured article with a well-detailed methodology. The authors discuss the feasibility and reproducibility of culture methods, aerosolization of viable and heat-inactivated A. versicolor, and lymphoproliferation of spores (the deposition and lymphoproliferative dose of A. versicolor) for use in challenge studies. They also performed an exposure study to examine lung responses to repeated inhalation of A. versicolor.

The importance of this article lies in the fact that it proposes a method of administration and aerosolization of conidia that imitates a natural exposure to the fungus, which allows the study of different mechanisms in the host-parasite interaction.

I have only one comment:

Regarding “the lymphoproliferative levels, where they observed that even at the highest dose of A. versicolor, it did not induce an immunomodulatory role as in the case of A. fumigatus”, the authors do not mention if they measured the virulence of this strain, since that this influences the stimulation of the immune response, which would also explain the high or low stimulation, depending on the virulence of A. versicolor.

Line 457: Change Aspergillus versicolor to A. versicolor

Line 463: Change Stachybotrys chartarum to S. chartarum

Line 508: Change Aspergillus fumigatus to A. fumigatus

Line 509: Change Stachybotrys chartarum to S. chartarum

 Uniform your references to the style of the journal.

Author Response

The authors sincerely thank Reviewer 1 for their comments, as well as the recommendations for changes in our manuscript. We have changed the abbreviations of genus names in the discussion section and corrected the references to use Chicago-MDPI style. We have also added language describing the virulence of A. versicolor. As the reviewer observed, the decreased virulence of A. versicolor, in comparison to A. fumigatus, is likely a factor driving the differences in lymphoproliferation. In future studies, we will consider investigating the unique components of each species and investigate how these individual components influence the observed responses. These items are discussed in lines 501-521 in the discussion suggestion. Thank you for the suggestions!

Reviewer 2 Report

Comments and Suggestions for Authors

This study presents the optimization and reproducibility of a protocol for administering A. versicolor conidia to rodents by inhalation through the nose only.

This study provides an experimental protocol that can be used in future studies to investigate the physiological effects of repeated pulmonary exposure to fungal conidia.

In the introduction, the authors should indicate if any data is known about the concentration of A. versicolor conidia in the environment. This information would be useful to understand how they determined the concentration of conidia they aerosolized in this study.

Lines 80-81 Add reference(s) that show the involvement of A. versicolor in allergic diseases

Conclusions are in agreement with the results obtained

References are appropriate, but must be presented in journal format

The tables and figures are adequate; they clearly illustrate the results presented

Author Response

The authors kindly thank Reviewer 2 for their review, as well as the recommendation to add references regarding the involvement of A. versicolor in allergic disease. We have added references to the sentence denoted on line 80. Additionally, we have corrected the references to be in Chicago-MDPI style. Thank you!

Reviewer 3 Report

Comments and Suggestions for Authors

This submission is well-written and has a number of important features aside from the data which is also very good. The manuscript is interdisciplinary, and includes experts in of Clinical Allery and Immunity, Pharma & Toxicology, Agrobiotechnology, Occupational Safety & Health, and Respiratory Health. The data describe a new mouse inhalation model that measures immune responses to Aspergillus versicolor, a major pathogen in this immunological disease. Immune data correlate with dosage and cell treatments.

I have a few comments on the data.

1.      The abstract, Introduction, Methods, and other sections of the paper are succinct. As for the methods, I am not familiar with all the details, but I assume they have been done correctly.

2.      Figure 1. I do not understand the contribution of Figure 1A and B, which compares heat-inactivated and not-inactivated cells. Data can be removed.  Was it included to show differences in cell morphology among both treatments? I believe the data of Figure 1, D-F reasonably shows the differences in treated versus untreated cells, although the data showed larger numbers of HIC cells (particles). Also, there is no HIC 7-day treatment particle data.

3.      Figure 2 A, there seems to be more protein in live versus HIC treatment. Is this significant?

4.      Figure 3 What is meant by a negative control? I see the attachment of HIC – treated cells. Explain please. In bronchioles there is no difference in cell treatments (HIC-treated cells no and non-treated viable cells.  Explanation? Also, in the alveoli, there is not much difference among treatments.  Is there supposed to be a difference?

5.      Figure 4 is fine.

6.      Figure 5.   A dose-dependency is verified.

7.      Why is data shown of Figure 7 first, than Figure 6? Should data be reversed?

Author Response

The authors kindly thank Reviewer 3 for their observations and comments on the manuscript. We would like to clarify that in this study, the heat-inactivated conidia (HIC) test article was used as a biological control. Therefore, we thought it was important to show that the heat treatment of the conidia did not alter morphology and that the morphology of the viable (live) conidia and the heat-inactivated conidia (HIC) is the same (Figure 1). However, in future studies utilizing fungal species, we will consider including the morphological data in the supplemental file.

In Figure 2, the reviewer observed that there were differences in some proteins. While the protein profiles depicted in the Western Blot look slightly different between the viable (live) protein lysate compared to the heat-inactivated protein lysate, the differences were only slight and not statistically significantly. In the future, we will consider investigating more deeply the proteomic profiles of different fungal species and the heat-inactivated control counterparts, which could provide insight to the unique immune responses to each of them.

In Figure 3, the reviewer asked for clarification on what was meant by negative control. The authors would like to clarify that the negative control was referenced in Figure 2 (C and D). This negative control was either sterile rice (Figure 2C) as a control for the measurement of sterigmatocystin as this measurement used fungal-laden rice, or a filter (Figure 2D) from an air-only control exposure pod to compared against filters from fungal-exposed pods housed on the exposure chamber. If referring to the control used in Figure 3, the heat-inactivated conidia (HIC) test article was used as a biological control. This figure illustrates that both the viable conidia and HIC are effectively deposited into the lung using the Acoustical Generating System. The reviewer is correct that the HIC are deposited into the bronchiole and alveoli similarly to the viable cells, though by quantification it was significantly less frequent. From these data, the authors conclude that any immunological differences downstream are not a result of the inability to aerosolize the conidia, but rather a result of viable vs heat-inactivated conidia.

Regarding Figures 6 and 7, Figure 6 is discussed first on lines 441-451, and assigned on line 452, and Figure 7 is discussed on likes 452-454, and assigned on line 455. We apologize if these were reversed in the document this reviewer received. Thank you for your comments and suggestions.